# Observability Analysis and Improvement Approach for Cooperative Optical Orbit Determination

**Yan Luo [1], Tong Qin [2],\* and Xingyu Zhou [1]**

[1] School of Aerospace Engineering, Beijing Institute of Technology, Beijing 100081, China; 3120190076@bit.edu.cn (Y.L.); zhouxingyu@bit.edu.cn (X.Z.)
[2] School of Information and Electronics, Beijing Institute of Technology, Beijing 100081, China
\* Correspondence: qintong@bit.edu.cn

**Abstract:** Cooperative orbit determination (OD) using inter-spacecraft optical measurements is an important technology for space constellation missions. In this paper, the observability of a two-spacecraft cooperative OD system is investigated. The influence of geometric configuration on the observability is analyzed, and two special unobservable configurations are identified. Then, an approach to improve the observability by involving an additional spacecraft is proposed. Comparative analysis of system observability shows that an extra spacecraft in the system could change the coplanar and symmetric configuration and improve the observability of the cooperative OD system. Monte-Carlo simulations are carried out, and results verify the observability improvement conclusion.

**Keywords:** cooperative orbit determination; optical measurements; observability analysis; orbit configuration

## 1. Introduction

Autonomous orbit determination (OD) is one of the basic technologies to ensure the completion of space missions. With the dramatic increase of the number of spacecraft and the development of space cooperative behaviors, such as space constellation and formation missions [1,2], OD based on inter-spacecraft measurement becomes an important feature [3–6]. The techniques for estimating the absolute orbits of both the observer and the target using the inter-spacecraft measurements, which is called the cooperative OD, are increasingly desired.

The work on cooperative OD can be traced back to the 1980s, when Markley and Psiaki [7] proposed the OD method for two spacecraft using inter-spacecraft relative position measurements. The relative position measurement includes both distance and direction information, which requires several types of sensors, such as a laser sensor measuring relative distance and an optical camera measuring the relative direction [8]. With the miniaturization development of spacecraft, the onboard sensors are limited [9–11]. Another two widely used methods for cooperative OD are based on range-only measurements [12–16] and line-of-sight (LOS) measurements [17–19]. Ranging information can be obtained through radio, radar, or laser, and thus range measurements are hardly affected by the complex space environment and low cost [20]. However, the lack of angle information inevitably leads to overall rotation of the constellation, and absolute OD is unavailable [21,22]. The LOS measurement can be easily obtained using optical cameras with the advantages of long observation arc, high measurement accuracy and long detection range [23–25]. Moreover, the optical measurement, including the right ascension and the declination [26,27], can provide more information than the relative range. Therefore, angle-only measurements are of great value in the cooperative OD problem.

Regarding the cooperative OD problem based on inter-spacecraft optical measurements, observability analysis is particularly important [28–32]. It has been demonstrated that an OD system including one observer and one target is observable using inertial LOS

measurements [33–35]. The inertial LOS measurements can be obtained by combining the relative LOS information and attitude of the observer in the inertial coordinate. The cooperative OD system using inertial LOS measurement is first analyzed by Yim et al. [33]. The conclusion that the two spacecrafts are observable was obtained through numerical simulations. Hu further investigated the observability of an inertial optical cooperative OD system with three spacecraft using the Lie-derivative criterion [34,35]. It was declared in Hu's work that the three-spacecraft system is third-order locally, weakly observable. However, the previous conclusion does not hold in some cases. Specifically, the inertial cooperative OD system is unobservable under certain orbital geometric configurations, and observability of the OD system is poor when configuration is close to those special cases. Thus, it is necessary to further investigate the influence of orbital configuration on observability of a cooperative OD system.

This paper investigates the observability of the cooperative optical OD system. Differently from the previous studies of this problem focusing on the measurements and dynamics, this paper handles the problem in view of the orbit configuration. The contributions of this paper are summarized as follows. Firstly, the observability of a two-spacecraft optical cooperative OD system with different configurations is analyzed based on the observability matrix (OM). Several unobservable configurations are dug out and the corresponding unobservable elements are discussed. Secondly, an observability improvement strategy for the unobservable configurations is proposed by adding an additional observer. Simulations are performed to verify the enhancement effect on observability, and OD accuracy is analyzed and compared in conditions with and without the additional observer.

This paper is organized as follows: In Section 2, the observability of the inertial optical cooperative OD is analyzed. In Section 3, the observability improvement method is proposed and studied. Numerical simulations are implemented for comparisons through the Monte-Carlo method for accuracy analysis.

## 2. Observability Analysis of the Two-Spacecraft OD System

### 2.1. System Description

2.1.1. State Model

The cooperative OD system includes two spacecraft orbiting around Earth, known as the observer spacecraft ($S_O$) and the target spacecraft ($S_T$), with the Keplerian orbit elements given as:

$$\boldsymbol{E}_O = [a_O, \, e_O, \, i_O, \, \omega_O, \, \Omega_O, \, n_O]$$
$$\boldsymbol{E}_T = [a_T, \, e_T, \, i_T, \, \omega_T, \, \Omega_T, \, n_T] \tag{1}$$

where the subscripts $O$ and $T$ represent the two spacecraft, $S_O$ and $S_T$, respectively. The observation spacecraft can actively observe the target with onboard camera, while the target spacecraft is passively observed and is usually a non-cooperative target. For each spacecraft, the orbit elements denote semi-major axis, eccentricity, inclination, argument of the periapsis, longitude of the ascending node, and true anomaly, respectively. The goal of a cooperative OD system is to determine the orbit elements of both $S_O$ and $S_T$. The system state vector $\boldsymbol{x}$ is given by:

$$\boldsymbol{x} = [a_O, \, e_O, \, i_O, \, \omega_O, \, \Omega_O, \, n_O, a_T, \, e_T, \, i_T, \, \omega_T, \, \Omega_T, \, n_T]^T \tag{2}$$

Considering point-mass two-body dynamics without any non-gravitational perturbation [36], the size, shape, and spatial position of each orbit described by the first five parameters of $\boldsymbol{E}_O$ and $\boldsymbol{E}_T$ are time-invariant [35–38]. The true anomaly, which represents the position of the spacecraft in orbit, is time varying. Then, the state model of the cooperative OD system is given by:

$$\dot{\boldsymbol{x}} = f(\boldsymbol{x}) = [\, 0, \, 0, \, 0, \, 0, \, 0, \, \dot{n}_O, \, 0, \, 0, \, 0, \, 0, \, 0, \, \dot{n}_T]^T \tag{3}$$

$$\begin{aligned}
\dot{n}_O &= \sqrt{\frac{\mu}{a_O\left(1-e_O^2\right)}}\,\frac{(1+e_O\cos(n_O))^2}{a_O\left(1-e_O^2\right)} \\
\dot{n}_T &= \sqrt{\frac{\mu}{a_T\left(1-e_T^2\right)}}\,\frac{(1+e_T\cos(n_T))^2}{a_T\left(1-e_T^2\right)}
\end{aligned} \tag{4}$$

### 2.1.2. Measurement Model

Space-based optical measurements use optical cameras to obtain images of the surrounding environment, and the measurement information is then passed through a signal processor to extract the target. For a distant space target, its image is often seen as a light spot on the image plane, so that we can obtain only the target's line-of-sight (LOS) information relative to the observation spacecraft. Combining the attitude of $S_O$ received via star sensors, the inertial LOS measurement can be obtained.

The inertial LOS measurement model is given as:

$$h(\boldsymbol{x}) = \frac{\boldsymbol{r}_T - \boldsymbol{r}_O}{\|\boldsymbol{r}_T - \boldsymbol{r}_O\|} + \boldsymbol{v} \tag{5}$$

where $\boldsymbol{r}_O$ and $\boldsymbol{r}_T$ represent the inertial position vectors of $S_O$ and $S_T$ at the time of measurement regardless of the time it takes for the light to travel, and $\boldsymbol{v}$ is the measurement noise. The LOS measurement is defined in the Earth centered inertial frame ($O_E - X_E Y_E Z_E$), and a schematic diagram of the measurement model is shown in Figure 1.

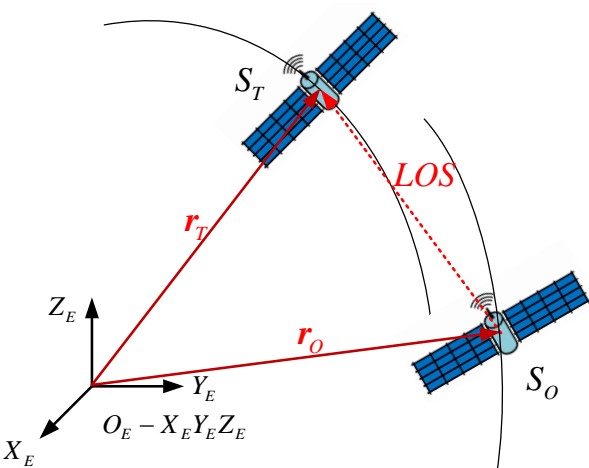

**Figure 1.** Orbits and inertial LOS Measurement.

To analyze the effect of orbital configurations, the LOS measurement should be expressed as a function of the system state variables, and we should describe the inertial position vectors by the orbit elements. The coordinate transformation is the same for each spacecraft, so the subscripts are omitted during the derivation of the inertial position vector.

The near-focal coordinate system ($O - \overline{xyz}$) describes the natural properties of an orbit [39], which takes the focus of the orbit (center of the earth for earth-orbiting spacecraft) as the origin, and the orbital plane is the reference plane. The $\bar{x}$-axis points to the perigee direction from the center of the earth, the $\bar{y}$-axis is obtained by rotating the $\bar{x}$-axis by $90°$ along the motion direction in the orbital plane, and the $O - \overline{xyz}$ system is a right hand coordinate system.

In the $O - \overline{xyz}$ system, the position vector of a spacecraft can be expressed by the orbit elements as follows:

$$\boldsymbol{r}_{\bar{x}} = \frac{a(1-e^2)}{1+e\cos n}\begin{bmatrix} \cos n \\ \sin n \\ 0 \end{bmatrix} \tag{6}$$

where the subscript $\overline{x}$ represents the $O - \overline{xyz}$ system, and $r_{\overline{x}}$ is the position vector in $O - \overline{xyz}$ system. The orthogonal transformation matrix from $O_E - X_E Y_E Z_E$ to $O - \overline{xyz}$ is:

$$Q_{X\overline{x}} = T_z(\omega)T_x(i)T_z(\Omega) \tag{7}$$

where $T_z$ and $T_x$ are Euler orientation cosine matrixes.

Then the transformation matrix from $O - \overline{xyz}$ to $O_E - X_E Y_E Z_E$ is the transpose of $Q_{X\overline{x}}$, that is:

$$Q_{\overline{x}X} = Q_{X\overline{x}}^T = \begin{bmatrix} \cos(\Omega) & \sin(\Omega) & 0 \\ -\sin(\Omega) & \cos(\Omega) & 0 \\ 0 & 0 & 1 \end{bmatrix} \begin{bmatrix} 1 & 0 & 0 \\ 0 & \cos(i) & \sin(i) \\ 0 & -\sin(i) & \cos(i) \end{bmatrix} \begin{bmatrix} \cos(\omega) & \sin(\omega) & 0 \\ -\sin(\omega) & \cos(\omega) & 0 \\ 0 & 0 & 1 \end{bmatrix} \tag{8}$$

The inertial position vector $r$ can be described by the orbit elements as:

$$r = Q_{\overline{x}X} r_{\overline{x}} = Q_{\overline{x}X} \frac{a(1-e^2)}{1+e\cos n} \begin{bmatrix} \cos n \\ \sin n \\ 0 \end{bmatrix} \tag{9}$$

2.1.3. Observability Matrix

The measurement model only reflects the relationship between the observation and the spacecraft state at a certain observation moment. The spacecraft state changes with time during the whole observation period, and its changing rule is described by the dynamic model. The observability matrix (OM) combines the measurement equation and the dynamics equation, and thus characterizes the relationship between observations and states over the entire observation period [18].

Using the method in [14], the OM of the two-spacecraft OD system, signed as $M$, is constructed based on the state equation in Equation (3) and the observation equation in Equation (5). The partial differential matrix of the measurement model at the observation time $t_i$ is given by:

$$H_i(x) = \left[ \frac{\partial h}{\partial x} \right]_i \tag{10}$$

For each observation time $t_i$, $H_i(x)$ is a 3 by 12 matrix, and it should be mapped to the initial epoch $t_0$ through the state transformation matrix (STM), given as:

$$\widetilde{H}_i(x) = H_i(x)\Phi(t_i, t_0) \tag{11}$$

where $\Phi(t_i, t_0)$ is the STM from $t_0$ to $t_i$, and given as:

$$\begin{cases} \dot{\Phi}(t_i, t_0) = \left[ \frac{\partial f(x_1)}{\partial x_1} \right]_i \Phi(t_i, t_0) \\ \Phi(t_0, t_0) = I_{12\times12} \end{cases} \tag{12}$$

In the cooperative OD system, the STM has the following format:

$$\Phi(t_i, t_0) = \begin{bmatrix} A_O & 0_{6\times6} \\ 0_{6\times6} & A_T \end{bmatrix} \tag{13}$$

where $A_k(k = O, T)$ is derived as:

$$A_k = \begin{bmatrix} 1 & 0 & 0 & 0 & 0 & 0 \\ 0 & 1 & 0 & 0 & 0 & 0 \\ 0 & 0 & 1 & 0 & 0 & 0 \\ 0 & 0 & 0 & 1 & 0 & 0 \\ 0 & 0 & 0 & 0 & 1 & 0 \\ \Phi_{a_k}^{n_k}(t_i, t_0) & \Phi_{e_k}^{n_k}(t_i, t_0) & 0 & 0 & 0 & \Phi_{n_k}^{n_k}(t_i, t_0) \end{bmatrix} \tag{14}$$

The elements in matrix $A_k$ shows the state transformations for specific orbit elements. For the time-invariant elements, the state transformations caused by themselves are equal to 1, showing as the diagonal elements in the first five rows. The state transformations caused by the irrelevant orbit elements are equal to 0. It is known from Equation (4) that $\dot{n}$ is relevant to $a$, $e$ and $n$ for each spacecraft, so the state transformations of $n_k$ are expressed in the differential equations as:

$$
\begin{aligned}
\dot{\boldsymbol{\Phi}}_{a_k}^{n_k}(t_i, t_0) &= \frac{\partial \dot{n}_k}{\partial a_k} + \frac{\partial \dot{n}_k}{\partial n_k} \boldsymbol{\Phi}_{a_k}^{n_k}(t_i, t_0), \quad \boldsymbol{\Phi}_{a_k}^{n_k}(t_0, t_0) = 0 \\
\dot{\boldsymbol{\Phi}}_{e_k}^{n_k}(t_i, t_0) &= \frac{\partial \dot{n}_k}{\partial e_k} + \frac{\partial \dot{n}_k}{\partial n_k} \boldsymbol{\Phi}_{e_k}^{n_k}(t_i, t_0), \quad \boldsymbol{\Phi}_{e_k}^{n_k}(t_0, t_0) = 0 \\
\dot{\boldsymbol{\Phi}}_{n_k}^{n_k}(t_i, t_0) &= \frac{\partial \dot{n}_k}{\partial n_k} \boldsymbol{\Phi}_{n_k}^{n_k}(t_i, t_0), \quad \boldsymbol{\Phi}_{n_k}^{n_k}(t_0, t_0) = 1
\end{aligned}
\tag{15}
$$

where $\boldsymbol{\Phi}_{a_k}^{n_k}(t_i, t_0)$ describes the state transformation of $n_k$ (marked by the superscript) caused by $a_k$ (marked by the subscript). Similarly, $\boldsymbol{\Phi}_{e_k}^{n_k}(t_i, t_0)$ and $\boldsymbol{\Phi}_{n_k}^{n_k}(t_i, t_0)$ represent the state transformation of $n_k$ caused by $e_k$ and $n_k$.

Then, the OM is updated with observation in the form of:

$$
\boldsymbol{M} = \begin{bmatrix} \widetilde{\boldsymbol{H}}_1(\boldsymbol{x}) \\ \widetilde{\boldsymbol{H}}_2(\boldsymbol{x}) \\ \vdots \end{bmatrix}
\tag{16}
$$

For each observation time $t_i$, $\widetilde{\boldsymbol{H}}_i(\boldsymbol{x})$ is also a 3 by 12 matrix. It can be seen that $\widetilde{\boldsymbol{H}}_i(\boldsymbol{x})$ makes up $\boldsymbol{M}$ over time, and each column of $\boldsymbol{M}$ corresponds to the orbital element.

Further mathematical analysis based on OM is implemented to investigate the observability of an OD system. In the first place, checking the rank condition of OM is necessary for qualitatively analyzing the observability of the system. A full rank OM means the OD system is observable, while when OM is not full rank, we can draw the conclusion that the system is not observable. The rank of OM represents the number of observable states or state combinations. Secondly, the quantitative analysis of system observability is important for checking the degree of system observability, which is generally based on the computation of the condition number (CN). CN is the ratio of the singular values of OM, given by:

$$
CN = \frac{\max(\sigma(OM))}{\min(\sigma(OM))}
\tag{17}
$$

A larger $CN$ indicates a less observable problem, and the estimation problem is ill-conditioned when its value exceeds a certain limit. An OD system is regarded as unobservable if the condition number is greater than $10^{16}$ [32].

### 2.2. Observability of the Angle-Only Cooperative OD System

In this subsection, the influence of geometric configuration on the observability of the two-spacecraft system is discussed, and two unobservable configurations, including the symmetric case and the same circular orbit case, are analyzed. Numerical simulations are implemented to verify the observability results for different orbit configurations.

The obstruction of Earth is considered to ensure continuous visibility between the two spacecraft, and more complex environmental factors such as light conditions are not considered in this paper. The target spacecraft with an orbital altitude of 5000 km and the observation spacecraft with the orbital altitude greater than 4000 km is adopted to obtain continuous observation. The nominal orbital elements for the following three cases are shown in Table 1.

**Table 1.** Nominal orbital elements for observability analysis.

| Spacecraft | $a$/km | $e$ | $i$/deg | $\Omega$/deg | $\omega$/deg | $n$/deg | Annotation |
|---|---|---|---|---|---|---|---|
| $S_{T1}$ | 11,378.137 | 0.01 | 45 | 94.8 | 199.0 | −54.13 | Elliptical orbit |
| $S_{T2}$ | | 0 | | | | | Circular orbit |
| $S_{O1}$ | 10,378.137 | 0.05 | 45.05 | 29.93 | 132.9 | −17.74 | General case |
| $S_{O2}$ | 11,378.137 | 0.01 | 30 | 94.8 | 199.0 | −54.13 | Symmetric case |
| $S_{O3}$ | 11,378.137 | 0 | 45 | 94.8 | 199.0 | −24.13 | Same circular case |

### 2.2.1. General Case

Assume that two spacecraft move on orbits with totally different elements that satisfy Equations (18)–(20):

$$\Omega_1 - \Omega_2 \neq 2k\pi \text{ or } i_1 \neq i_2 \tag{18}$$

$$\Omega_1 - \Omega_2 \neq (2k-1)\pi \text{ or } i_1 \neq -i_2 \tag{19}$$

$$a_T \neq a_O, \ e_T \neq e_O, \ e_T \neq 0, \ e_O \neq 0 \tag{20}$$

where $k$ is an integer. Equations (18) and (19) mean that the two orbits are not coplanar, and Equation (20) reflects that they have different sizes and shapes. The orbits are shown in Figure 2.

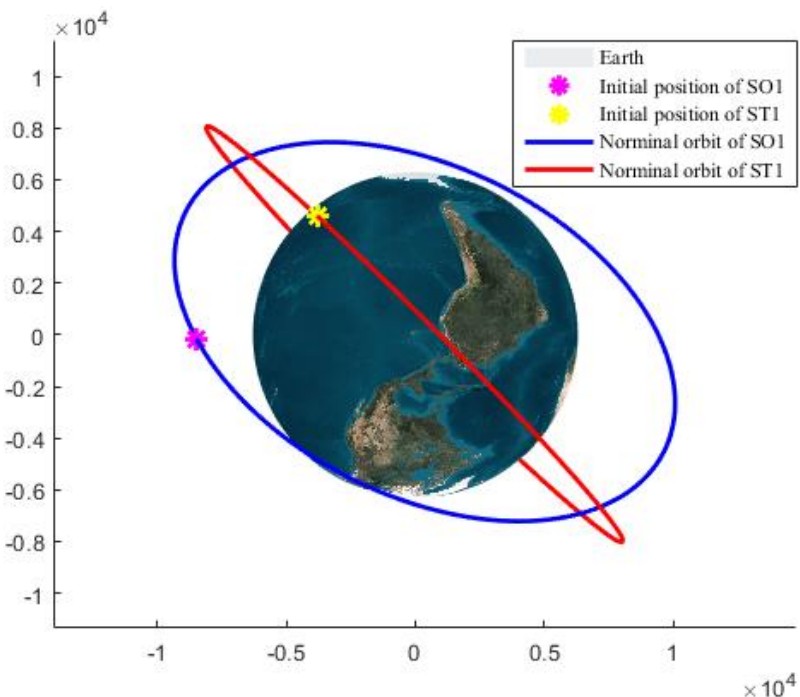

**Figure 2.** Orbits in general orbit configuration case.

In this case, the rank of the OM is calculated as 12, illustrating that all states in Equation (3) are observable, and the OD system is observable. The simulations of the OD problem are implemented by using the unscented Kalman filter (UKF). For each spacecraft in the OD system, the initial deviations of the triaxial position and triaxial velocity are set as 10 km and 1 m/s, respectively. Then, the initial covariance of state is defined as a diagonal matrix whose elements are the squares of the initial deviations. Then, the initial deviations of the orbital elements are obtained by converting state vectors to the six orbital elements, and the numerical values are shown in Table 2.

**Table 2.** Initial deviations of orbital elements.

| Spacecraft | *a*/km | *e* | *i*/deg | $\Omega$/deg | $\omega$/deg | *n*/deg |
|---|---|---|---|---|---|---|
| $S_{O1}$ | 33.3 | $1.3 \times 10^{-3}$ | $8.5 \times 10^{-5}$ | $1.2 \times 10^{-3}$ | $7.7 \times 10^{-3}$ | 0.039 |
| $S_{T1}$ | 18.6 | $2.7 \times 10^{-4}$ | $1.1 \times 10^{-4}$ | $1.6 \times 10^{-4}$ | 0.015 | 0.129 |

The standard deviation of measurement noise is set to 0.01 deg for LOS measurements according to the payload capabilities [40,41]. The process noise of the UKF is set to $10^{-12}$, and the simulation step is 60 s, depending on the measurement frequency. The initial simulation parameters are shown in Table 3.

**Table 3.** Simulation parameters used in the UKF.

| Name | Value |
|---|---|
| Initial state deviation of each spacecraft (km, km/s) | $[10, 10, 10, 1 \times 10^{-3}, 1 \times 10^{-3}, 1 \times 10^{-3}]^{\mathrm{T}}$ |
| Initial covariance of each spacecraft (km², km²/s²) | diag $([100, 100, 100, 1 \times 10^{-6}, 1 \times 10^{-6}, 1 \times 10^{-6}])$ |
| Standard deviation of measurement noise | 0.01 deg (approximately 40 arcsec) |
| Process noise | diag $(10^{-12})$ |
| Simulation step | 60 s |

Taking the case in which $S_{O1}$ observes $S_{T1}$ as a numerical example, the errors of orbit elements are shown in Figure 3. The estimation errors converge within four hours, and the sub-windows show the local magnifications of the estimation errors within 8–12 h. The estimation accuracies of semi-major axes are within 0.1 km, the accuracies of eccentricities are within $3 \times 10^{-5}$, the accuracy of *i* reaches the level of $10^{-5}$, the estimated accuracies of $\Omega$, $\omega$ and *n* reach the level of $10^{-4}$.

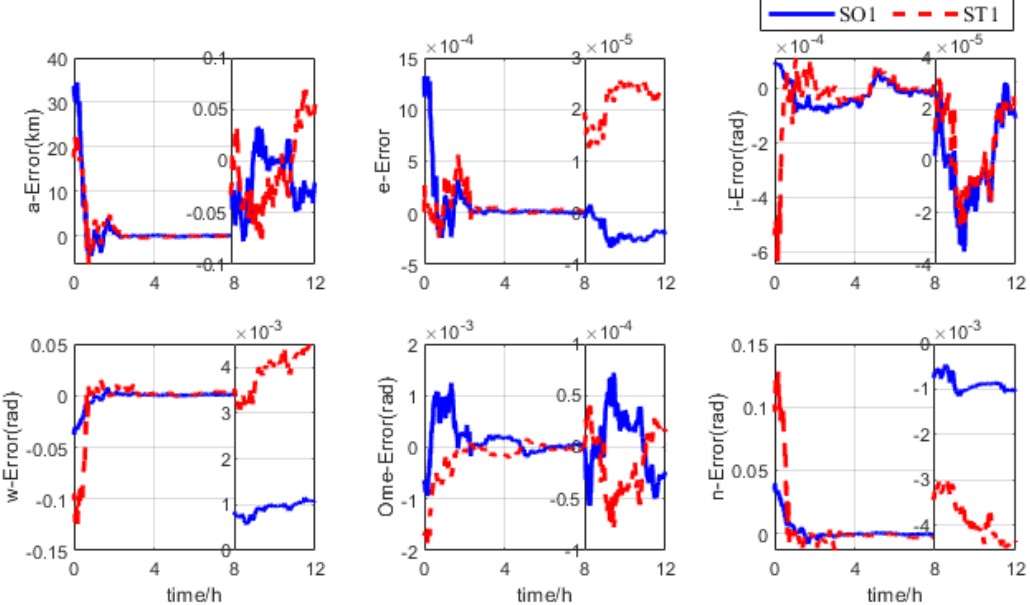

**Figure 3.** Estimation errors of orbit elements in general case.

The orbit determination precisions are obtained through Monte-Carlo simulations [42]. The final standard deviations (STDs) and convergency ratio (CR) of the $S_T$ are shown in Table 4.

**Table 4.** Final STD and CR results for the two-spacecraft OD system in general cases.

| Index | x/km | y/km | z/km | vx/(km/s) | vy/(km/s) | vz/(km/s) |
|---|---|---|---|---|---|---|
| STD | 0.0306 | 0.0200 | 0.0993 | $3.074 \times 10^{-5}$ | $2.9191 \times 10^{-5}$ | $2.4185 \times 10^{-5}$ |
| CR | 99.69% | 99.80% | 99.01% | 96.93% | 97.08% | 97.85% |

It can be seen that the states of target spacecraft can be determined in a general case. The STDs for triaxial position errors are within 0.1 km, and the convergency ratios are higher than 99%. The STDs for triaxial velocity errors are within 0.1 m/s, and the convergency ratios are higher than 96%.

### 2.2.2. Symmetric Case

In this case, the positions of the two spacecraft always remain symmetric with a stationary inertial plane. The restrictions in Equations (21) and (22) should be satisfied:

$$a_T = a_O, \ e_T = e_O, \ i_T \neq i_O, \ n_T = n_O \tag{21}$$

$$\begin{cases} \omega_T = \omega_O \\ \Omega_T = \Omega_O \end{cases} \text{or} \begin{cases} |\omega_T - \omega_O| = \pi \\ |\Omega_T - \Omega_O| = \pi \end{cases} \tag{22}$$

Equation (21) shows that the two orbits have the same semi-major axis and eccentricity, but different orbital inclinations. The restrictions of $\omega$ and $\Omega$ in Equation (22) indicate that the two orbits are mirror image symmetric in inertial space, and the same $n$ means that the two spacecraft are symmetric with the same inertial plane.

Let $S_{O2}$ observe $S_{T1}$, all orbit elements of the two orbits are different except for the inclinations. Numerical analysis shows that the rank of OM decreases to 6, meaning that there are 6 observable elements or element combinations. The OM can be simplified to the reduced row echelon form through Gauss–Jordan elimination method. Only six rows of the simplified matrix are valid, and the elements in the remaining rows are zero. The first six rows are shown in Table 5, and the numerical values that are not listed in the table are all equal to 0.

**Table 5.** The reduced row echelon in symmetric case.

| Row | $a_O$ | $e_O$ | $i_O$ | $\omega_O$ | $\Omega_O$ | $n_O$ | $a_T$ | $e_T$ | $i_T$ | $\omega_T$ | $\Omega_T$ | $n_T$ |
|---|---|---|---|---|---|---|---|---|---|---|---|---|
| 1 | 1 | | | | | | −1 | | | | | |
| 2 | | 1 | | | | | | −1 | | | | |
| 3 | | | 1 | | | | | | 1 | | | |
| 4 | | | | 1 | | | | | | −1 | −1.3 | |
| 5 | | | | | 1 | | | | | | 0.92 | |
| 6 | | | | | | 1 | | | | | | −1 |

Each row of the simplified matrix has more than one number, indicating that no independent element is observable, and there are six pairs of observable element combinations. Taking the first row as an example, there are two numbers in the columns that are corresponding to $a_O$ and $a_T$, showing that the combination $(a_O - a_T)$ is observable. Similarly, the 2nd, 3rd and 6th rows indicate that the combinations $(e_O - e_T)$, $(i_O + i_T)$ and $(n_O - n_T)$ are observable. The numbers in the 4th and 5th row show that the remaining angles, $\Omega_O$, $\Omega_T$, $\omega_O$ and $\omega_T$, make up two observable combinations in an irregular form.

The estimation errors of orbit elements over time are shown in Figure 4. It can be known that none of the error's convergence, which verifies the observability conclusion that no independent element is observable. The estimation errors of the combinations $(a_O - a_T)$, $(e_O - e_T)$, $(i_O + i_T)$ and $(n_O - n_T)$ in Figure 5 are convergent, which proves the observability results for orbit element combinations.

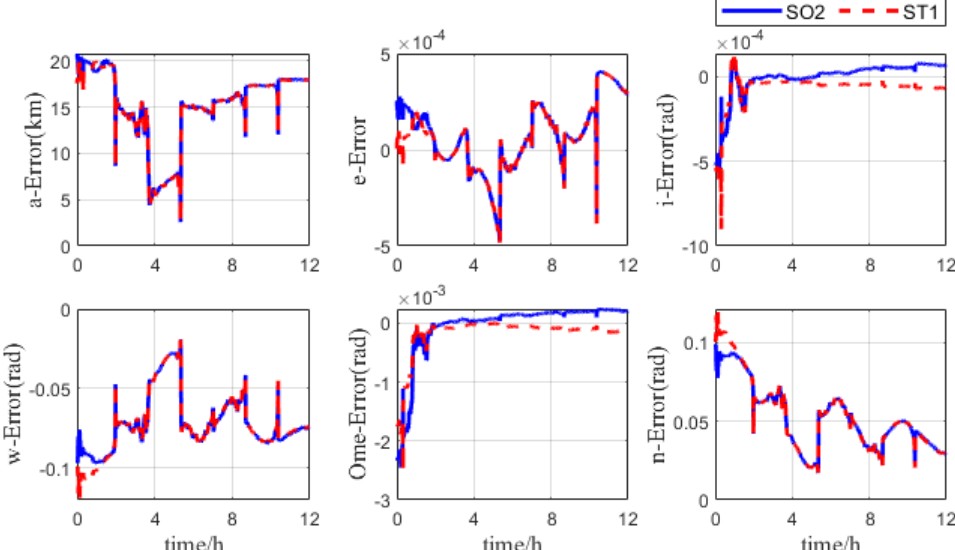

**Figure 4.** Estimation errors of orbit elements in the symmetric case.

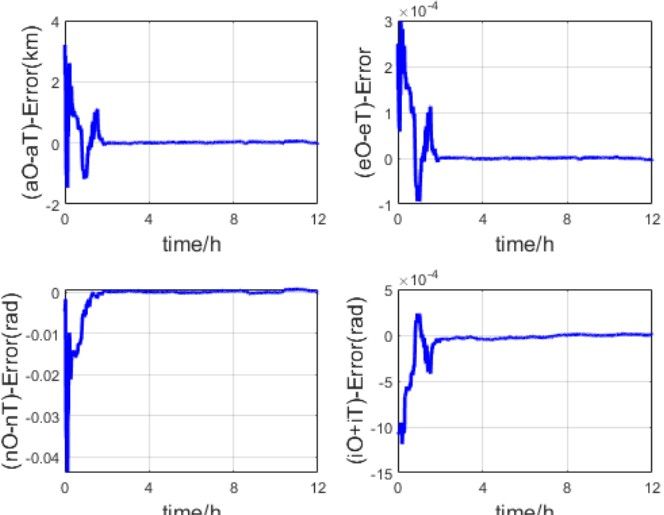

**Figure 5.** Estimation errors of orbit element combinations in the symmetric case.

From the point of view of OD, this symmetrical configuration is unfavorable. The LOS direction is always perpendicular to the symmetry plane of the two spacecraft, and therefore doesn't contain the information related to the semi-major axis, eccentricity and the true anomaly. Moreover, the LOS vectors before and after the cross epoch are opposed to each other. In this case, the drastic changes of the measurements cause the estimations to diverge (as shown in Figure 4). Although this symmetric case rarely appears in practice, it is worth investigating because the observability and convergence will also be poor in configurations close to the symmetric case.

### 2.2.3. Same Circular Orbit Case

For two spacecraft on the coplanar orbits, the elements should satisfy one of the following equations:

$$i_1 = i_2 \text{ and } \Omega_1 - \Omega_2 = 2k\pi \tag{23}$$

$$i_1 = -i_2 \text{ and } \Omega_1 - \Omega_2 = (2k-1)\pi \tag{24}$$

In this case, the two spacecraft move on the same circular orbit with only different true anomalies, and the remaining elements should satisfy:

$$a_T = a_O, \; e_T = e_O = 0, \; n_T \neq n_O \tag{25}$$

Let $S_{O3}$ observe $S_{T2}$. The rank of OM becomes 6, and the reduced echelon form is shown in Table 6. For the 2nd and 6th row, there is only one number in each row, indicating that the corresponding elements, $e_O$ and $e_T$, are independently observable. The numbers corresponding to the column of $a_T$ are in the 1st and 4th row. There is another number corresponding to $a_O$ in the 1st row, and there are 7 numbers in the 4th row. Meanwhile, the numbers corresponding to $i_T$ and $\Omega_T$ also appear in the 3rd, 4th and 5th rows. Therefore, the other 4 observable variables are composed of the semi-major axes and the eight angles in a combinatorial form.

**Table 6.** The reduced row echelon in same circular orbit case.

| Row | $a_O$ | $e_O$ | $i_O$ | $\omega_O$ | $\Omega_O$ | $n_O$ | $a_T$ | $e_T$ | $i_T$ | $\omega_T$ | $\Omega_T$ | $n_T$ |
|---|---|---|---|---|---|---|---|---|---|---|---|---|
| 1 | 1 | | | | | | 1 | | | | | |
| 2 | | 1 | | | | | | | | | | |
| 3 | | | 1 | | | | | | $-0.866$ | | $-0.354$ | |
| 4 | | | | 1 | | $-1$ | $-6.5 \times 10^{-4}$ | | $-0.5$ | 1 | 1.319 | $-1$ |
| 5 | | | | | 1 | | | | 0.7071 | | $-0.866$ | |
| 6 | | | | | | | | 1 | | | | |

The estimation errors of orbit elements over time are shown in Figure 6. It can be seen that the errors of $a$, $i$, $\omega$, $\Omega$ and $n$ are not convergent, proving that the 10 elements are unobservable. For the errors of $e$, there are tendencies of convergence, but the accuracies are not high due to the influence of other orbit elements.

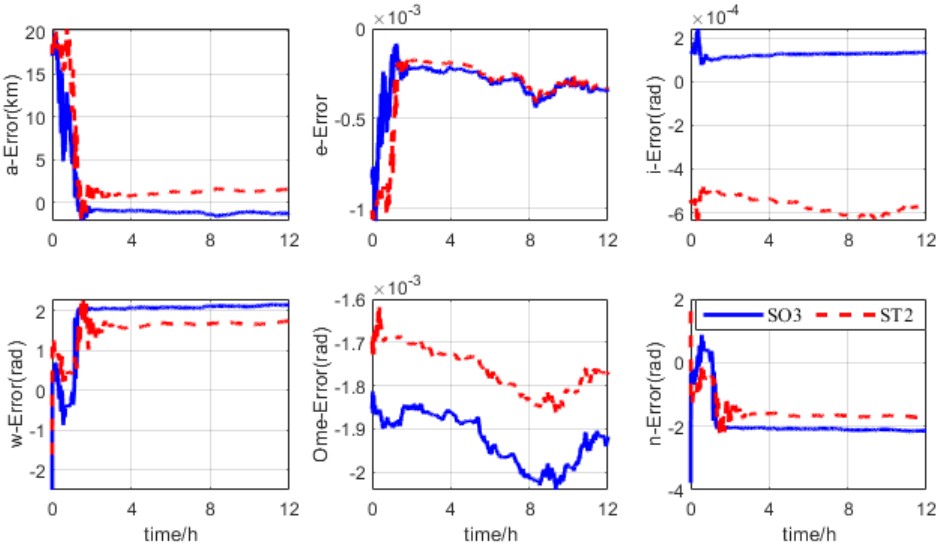

**Figure 6.** Estimation errors of orbit elements in the same circular orbit case.

## 3. Improvement Approach for Cooperative Optical Orbit Determination

It can be drawn from the previous section that the observability of a two-spacecraft cooperative OD system is limited by the relative geometric configuration. To solve this problem, an observability improvement approach is proposed by adding an additional observer into the cooperative OD system to obtain more measurement information. In this section, the optical cooperative OD system with an additional observer is modeled, and the system observability is analyzed.

### 3.1. System Description for the Cooperative OD System with an Additional Observer

In this cooperative OD scenario, each observation spacecraft measures the LOS from itself to the target spacecraft. The measurement model is given as follows:

$$h' = \left[ h_1^T \, h_2^T \right]^T \tag{26}$$

where $h_1$ and $h_2$ are the LOS measurements taken by the two-observation spacecraft. The measurement model is shown in Figure 7, $S_{O1}$ and $S_{O2}$ are observation spacecraft.

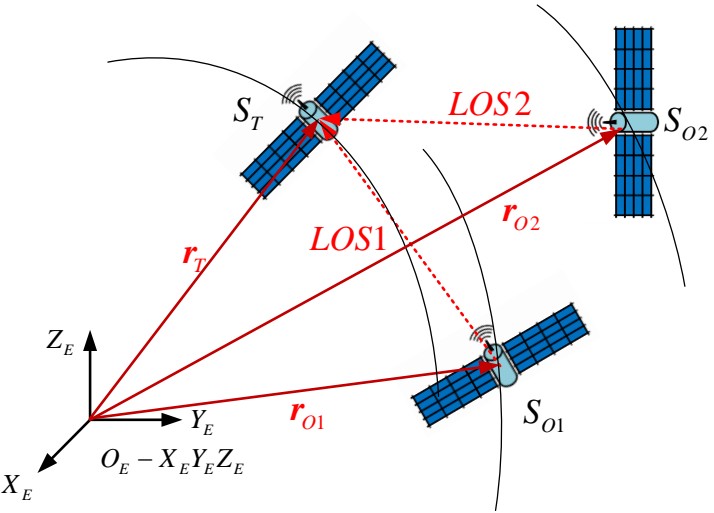

**Figure 7.** LOS measurement model for a cooperative OD system with an additional observer.

The elements of the target and the two observers should be estimated, and there are six orbital elements for each spacecraft. Then the state variables become:

$$x' = [E_{O1}, \, E_{O2}, \, E_T]^T \tag{27}$$

The OM of the cooperative OD system with an additional observer, defined as $M'$, is constructed in the same way as in Equations (10)–(16), with the observation matrix $H_i(x')$ and the state transformation matrix $\Phi'$ described as:

$$H_i(x') = \left[ \frac{\partial h'}{\partial x'} \right]_i \tag{28}$$

$$\Phi'(t_i, t_0) = \begin{bmatrix} A_{O1} & 0_{6\times6} & 0_{6\times6} \\ 0_{6\times6} & A_{O2} & 0_{6\times6} \\ 0_{6\times6} & 0_{6\times6} & A_T \end{bmatrix} \tag{29}$$

where the matrix $A_k(k = O1, \, O2, \, T)$ has the same form shown in Equations (14)–(16).

Then, we can get the intermediate matrix $\widetilde{H}_i(x')$, and $M'$ is given as:

$$M' = \begin{bmatrix} \widetilde{H}_1(x') \\ \widetilde{H}_2(x') \\ \vdots \end{bmatrix} \tag{30}$$

Similarly, the 18 columns of $M'$ correspond to the 18 orbital elements.

### 3.2. Observability Analysis for the Cooperative OD System with an Additional Observer

With an additional spacecraft involved in the OD system, not only the measurement information, but also the number of states, are increased. The observability of the system

requires further analysis. In this part, three cases corresponding to the configurations in Sec. II are analyzed to show the observability improvement. The orbital elements of the additional spacecraft, signed as $S_{O4}$, are shown in Table 7.

**Table 7.** Nominal orbital elements for observability analysis.

| Spacecraft | *a*/km | *e* | *i*/deg | $\Omega$/deg | $\omega$/deg | *n*/deg |
|:---:|:---:|:---:|:---:|:---:|:---:|:---:|
| $S_{O4}$ | 11,878.137 | 0.02 | 30 | 0 | 0 | 10 |

### 3.2.1. General Case

The first scenario considers two general non-coplanar observers. According to the observability result of the general case in a two-spacecraft system, either of the observation spacecraft can make up an observable OD system with the target spacecraft, so the cooperative OD system with both observers is completely observable. Therefore, it is feasible to estimate all orbital elements, although the dimension of the state variables increases.

Let $S_{O1}$ and $S_{O4}$ observe $S_{T1}$, and the orbits are shown in Figure 8. The rank of OM is calculated to be 18, as expected, and the observability of cooperative OD system is enhanced, reflected in the fact that the numerical value of *CN* decreases to $6.7037 \times 10^3$. The errors in Figure 9 show that the accuracies of semi-major axes are around 20 m, the estimated errors of eccentricities reach the level of $10^{-6}$, the estimated accuracies of *i* and $\Omega$ reach the level of $10^{-6}$, and the estimated accuracies of $\omega$ and **n** reach the level of $10^{-4}$.

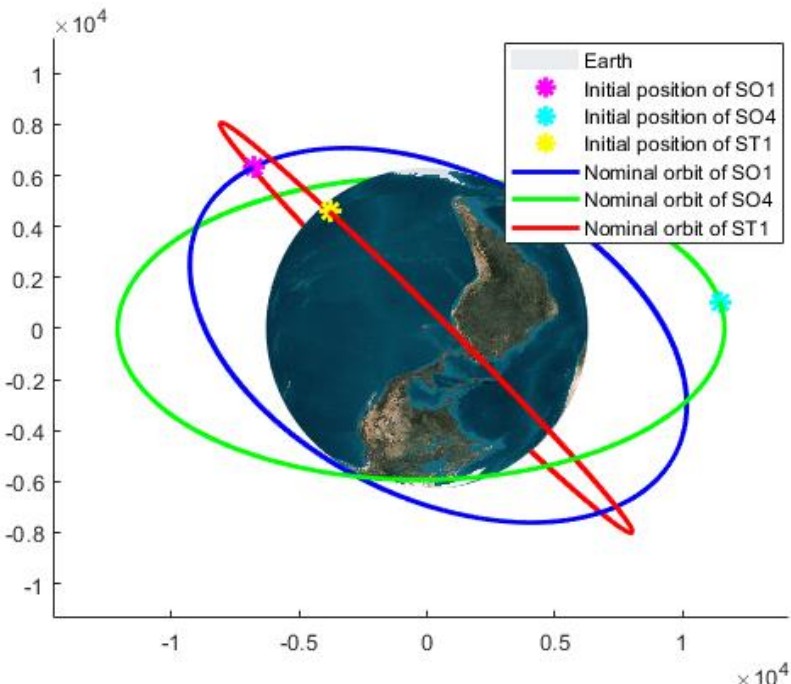

**Figure 8.** Orbits in general orbit configuration case with an additional observer.

To validate the advantage of using an additional observer in the inertial LOS cooperative OD system, the OD accuracies of the three-spacecraft system proposed are analyzed. For the general case with two general observation spacecraft, the STD and CR results of 100 case Monte-Carlo simulations are shown in Table 8. Comparing with the Monte-Carlo results for two-spacecraft system in Table 4, the values of STDs are much smaller. The STDs for triaxial position errors are within 0.015 km, and the STDs for triaxial velocity errors are within 0.01 m/s. The convergency ratios are higher than 99%.

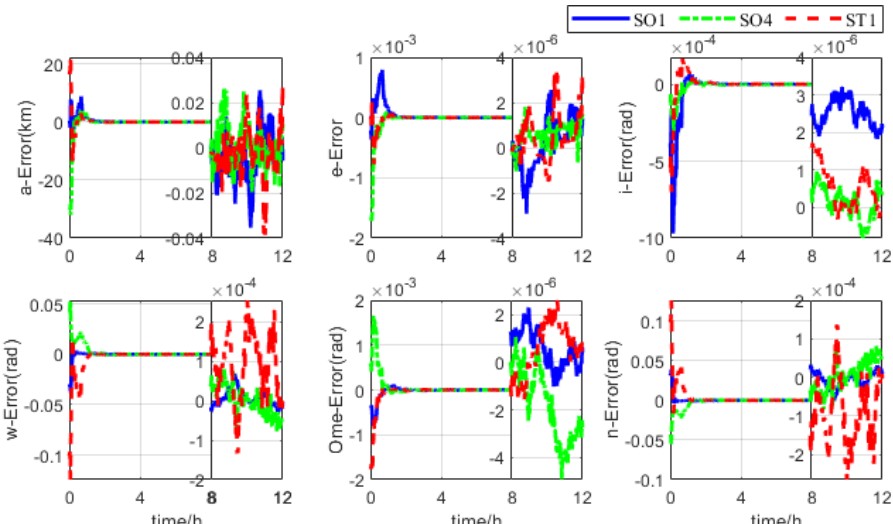

**Figure 9.** Errors of orbit elements in general case.

**Table 8.** Final STD and CR results for the cooperative OD with an additional observer.

| Configuration | Index | x/km | y/km | z/km | vx/(km/s) | vy/(km/s) | vz/(km/s) |
|---|---|---|---|---|---|---|---|
| General orbit | STD | 0.0091 | 0.0010 | 0.0117 | $1.1744 \times 10^{-6}$ | $6.0360 \times 10^{-6}$ | $9.2720 \times 10^{-7}$ |
| | CR | 99.99% | 99.99% | 99.99% | 99.88% | 99.40% | 99.99% |
| Symmetric orbit | STD | 0.0018 | 0.0076 | 0.113 | $5.2919 \times 10^{-6}$ | $5.0495 \times 10^{-6}$ | $7.3012 \times 10^{-6}$ |
| | CR | 99.99% | 99.99% | 99.99% | 99.47% | 99.50% | 99.27% |
| Same circular orbit | STD | 0.0036 | 0.0060 | 0.0105 | $5.4042 \times 10^{-6}$ | $1.3093 \times 10^{-6}$ | $4.3750 \times 10^{-6}$ |
| | CR | 99.99% | 99.99% | 99.89% | 99.46% | 99.87% | 99.56% |

To compare the OD accuracies with and without the additional observer, the root-mean-square error (RMSE) of Monte-Carlo simulations are shown in Figure 10. The top two images are the RMSEs of the case without additional observer (i.e., two-spacecraft case), and the lower subgraphs show the RMSEs of the case with additional observer (i.e., three-spacecraft case). For the two-spacecraft case, the RMSEs of the position are smaller than 1.5 km, and the RMSEs of the velocity are smaller than $8 \times 10^{-4}$ km/s. For the three-spacecraft case, the RMSEs of position are within 0.1 km, and the RMSEs of velocity are within $5 \times 10^{-5}$ km/s. The OD accuracies of the three-spacecraft system are better. Moreover, the estimated errors of the three-spacecraft OD system converge after about 2 h, which are much faster than in the two-spacecraft system.

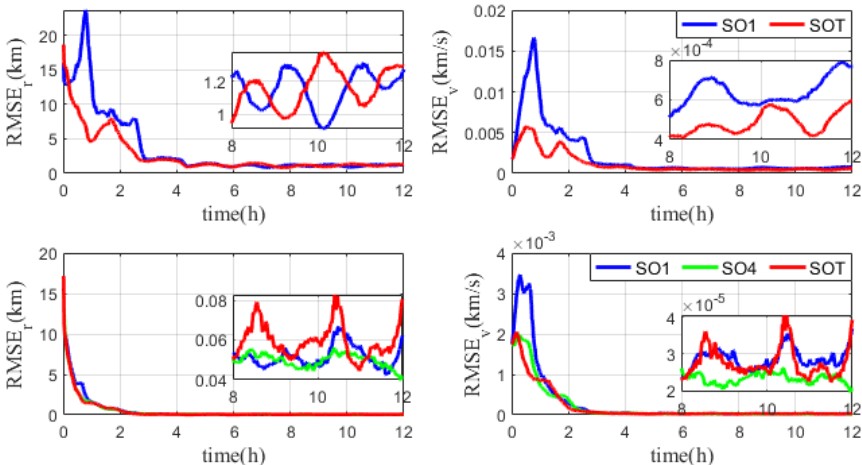

**Figure 10.** RMSE results of Monte-Carlo simulations.

### 3.2.2. Symmetric Case

It has been proven that the cooperative OD system is unobservable with only an observer on the symmetric orbit with the target, and neither of the two orbits can be determined. Let $S_{O2}$ and $S_{O4}$ observe $S_{T1}$. The OM is calculated to be full rank, meaning that the system is observable and even the orbit of the symmetric observer is available. The cooperative OD system determines the orbits of $S_{O1}$ and $S_{T1}$ firstly, as in the general OD case, and then determines the orbit of $S_{O3}$ based on this reference. The effect of symmetric configuration has been diminished but not eliminated, and the value of CN in this case is an order of magnitude larger than that in the general case, indicating that the system is still observable, but with weaker observability. The errors of orbital elements in Figure 11 show that the element accuracies can reach the same level, as in general case with an observation time longer than 3 h. It should be noted that the additional observer cannot be in special configurations, because the combining of singular cases is not able to break the previous unobservable configurations.

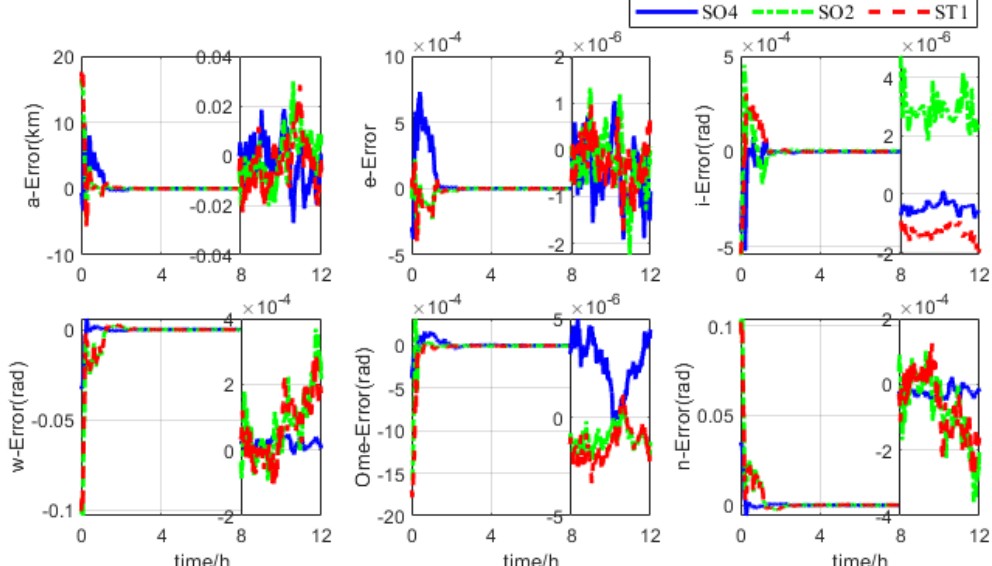

**Figure 11.** Errors of orbit elements in symmetric case.

The OD accuracy results for the system with a symmetric observer and a general observer are shown in Table 8. It can be seen that the accuracies reach the same level as the case with two general observers.

### 3.2.3. Same Circular Observer Case

Let $S_{O3}$ and $S_{O4}$ observe $S_T$. The value of CN increases sharply and reaches the level of $10^{16}$. The rank of OM is calculated to be 16, while the unobservable situation is caused by the circular orbits, which leads to the combination of $(\omega_{O3} + n_{O3})$ and $(\omega_{T2} + n_{T2})$ observable instead of the four individual elements. Therefore, it is proven that adding a general observer can make the same circular orbits observable.

The advantage of including two observers is that the observability of the OD system improves. The orbit configuration between each observation spacecraft and the target is different, so that the system can obtain more effective information. The analysis above shows that the combination of various orbital configurations can enrich measurement information to change the unobservable configurations, and thus improve the system observability. Same as in symmetric case, an additional observer on the same circular orbit or symmetric orbit with the target is not able to make the unobservable cooperative OD system observable.



## 4. Conclusions

In this paper, the observability of cooperative orbit determination system was analyzed based on numerical calculation of OM and analysis of the convergence of the orbit elements. The general case where the two spacecraft are on orbits with different elements is proven to be observable, while two unobservable orbit configurations are identified: (1) in the symmetric case, no independent element but six pairs of element combinations are observable; (2) in the same circular orbit case, there exist two observable individual elements and four element combinations. Involving an additional spacecraft has a remarkable effect on changing unobservable orbital configurations, and thus makes the OD system observable. For general cases, the numerical value of condition number decreases from the level of $10^6$ in two-spacecraft system to the level of $10^3$ in a three-spacecraft system. In terms of OD accuracy, the final STDs for triaxial position errors decreased from 0.1 km in a two-spacecraft system to within 0.015 km in a three-spacecraft system. The simulations show that an additional observer can improve the observability and OD accuracy of the cooperative OD system.

**Author Contributions:** Conceptualization, T.Q.; methodology, Y.L.; software, X.Z. All authors have read and agreed to the published version of the manuscript.

**Funding:** This research was funded by The National Natural Science Foundation of China, grant number 51827806, and The National Defense Basic Scientific Research Project, grant number JCKY2020903B002.

**Institutional Review Board Statement:** Not applicable.

**Informed Consent Statement:** Not applicable.

**Data Availability Statement:** Not applicable.

**Conflicts of Interest:** The authors declare no conflict of interest.

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
