# Peer review of "Observability Analysis and Improvement Approach for Cooperative Optical Orbit Determination"

_aerospace, doi:10.3390/aerospace9030166_

Round 1

Reviewer 1 Report

This paper first analyzes a two-spacecraft cooperative optical orbit determination system. Observability analysis shows the limitation of the two-spacecraft OD system. Then, an improvement approach is proposed and investigated using an additional observer. However, there are several problems in the Introduction and Analysis sections. I suggest accepting after revision.

Major issues:

1) pp. 2: Clearer explanations are needed in the Introduction section for the motivation, contributions of this paper, and gap of past literature. Two unobservable orbit configurations are depicted as results of this paper in the Conclusion. However, those results/conclusions are first provided in the Introduction section, which is not appropriate. Revisions are required for the Introduction.

2) Sec. 3.2: observability analysis in Sec. 3.2 shows the improvement of adding an additional observer. However, Table 7 shows that this additional observer is a "General case" to the target satellite. Then, it seems obvious that the main contribution of the improvement comes from the existence of a "General case" rather than having two observers. What if both observers are singular cases to the target? More analysis or at least the limitations of the proposed improvement approach should be provided.

3) pp. 18 & 19, results in Table 9 cannot lead to the conclusion of "the RMSE results in Table 9 show that the error convergences in the system with an additional observer are much faster than in two-spacecraft OD system". There is no information on the convergence speeds provided and no comparable data provided for three-spacecraft and two-spacecraft OD systems.

Minor issues:

1) pp. 5, Eq. (15), in the expression of Φeknk_dot, both sides have the term Φeknk_dot, which is a typo.

2) pp. 5, line 154, it is better to start a paragraph with a complete sentence.

3) pp. 7, Table 2 & Table 3, "1.6×10-4", "1×10-3".... Please keep the format consistent with other numbers.

4) pp. 7, it'll be better to also write the OM out like Table 4 & Table 5 for the general case so that it is easier for readers to compare the differences.

5) pp. 9, Fig. 6, please keep the format consistent with other figures for the labels of the x-axis. "Time/h" -> "time/h".

6) pp. 12, line 298, "3.1.1" should be "3.2.1", the same mistakes exist for the following numberings.

7) Sec. 3.2: it'll be better to also provide figures for the orbit configurations, as provided in Sec. 2.2 (e.g., Figs. 2, 4, & 7), which makes the problem more illustrative to readers.

8) pp. 14, line 329: is this "ST1" or should it be "ST2"?

9) pp. 14, line 330: typo, "1016"

Author Response

The authors particular appreciate the attention of the editors and reviewers on the manuscript. Each piece of comment is considered as a promotion on the overall quality of this article. All questions have been addressed carefully according to the advice. The specific responses and changes are listed in the Word. Please see the attachment.

Reviewer 2 Report

The authors described an orbit determination problem based on inter-spacecraft observables. The paper deals with a simplified model based on Earth-orbiting satellites. The method used is not innovative but the results can be interesting to a general audience. 

I attach my detailed comments in the attached document, and I add here my general comments:

1) Some sentences (see the attached document) must be clarified since their meaning is not clear. 

2) the results section must be shortened and made more clear in both paragraphs 2 and 3. 

3) the authors use too many images not adding much to the paper results. 

4) Paragraphs 3.3 and subsections must be summarized and included directly in the analysis of the specific and relevant configuration. 

5) conclusions must report some numerical result of the analysis. 

Author Response

(The authors gave the same response as above.)

Round 2

Reviewer 2 Report

The author improved the quality of the manuscript with respect to the first version. 

I would recommend reducing the number of tables (12) and figures (16). These are way too many. This also makes it difficult to understand a large number of results presented. 

From section 2.2, the number of results presented is too large and makes it difficult to appreciate the most important aspects of the paper. 

Author Response

The authors particular appreciate the attention of the editors and reviewers on the manuscript. All questions have been addressed carefully according to the advice. Please see the attachment for specific responses and modifications in the revised version.
